# Improving the Inhibitory Effect of Phages against *Pseudomonas aeruginosa* Isolated from a Burn Patient Using a Combination of Phages and Antibiotics

**DOI:** 10.3390/v13020334

**Published:** 2021-02-21

**Authors:** Bahareh Lashtoo Aghaee, Mohammadali Khan Mirzaei, Mohammad Yousef Alikhani, Ali Mojtahedi, Corinne F. Maurice

**Affiliations:** 1Department of Microbiology, Faculty of Medicine, Hamadan University of Medical Sciences, Hamadan 65178-38678, Iran; b.aghaee@hotmail.com; 2Institute of Virology, Helmholtz Center Munich and Technical University of Munich, 85764 Neuherberg, Germany; m.khanmirzaei@helmholtz-muenchen.de; 3Department of Microbiology & Immunology, Faculty of Medicine and Health Sciences, McGill University, Montreal, QC H3G 0B1, Canada; 4Department of Microbiology, School of Medicine, Guilan University of Medical Sciences, Rasht 41938-33697, Iran

**Keywords:** *Pseudomonas aeruginosa*, antibiotic resistance, bacteriophage

## Abstract

Antibiotic resistance causes around 700,000 deaths a year worldwide. Without immediate action, we are fast approaching a post-antibiotic era in which common infections can result in death. *Pseudomonas aeruginosa* is the leading cause of nosocomial infection and is also one of the three bacterial pathogens in the WHO list of priority bacteria for developing new antibiotics against. A viable alternative to antibiotics is to use phages, which are bacterial viruses. Yet, the isolation of phages that efficiently kill their target bacteria has proven difficult. Using a combination of phages and antibiotics might increase treatment efficacy and prevent the development of resistance against phages and/or antibiotics, as evidenced by previous studies. Here, in vitro populations of a *Pseudomonas aeruginosa* strain isolated from a burn patient were treated with a single phage, a mixture of two phages (used simultaneously and sequentially), and the combination of phages and antibiotics (at sub-minimum inhibitory concentration (MIC) and MIC levels). In addition, we tested the stability of these phages at different temperatures, pH values, and in two burn ointments. Our results show that the two-phages-one-antibiotic combination had the highest killing efficiency against the *P. aeruginosa* strain. The phages tested showed low stability at high temperatures, acidic pH values, and in the two ointments. This work provides additional support for the potential of using combinations of phage–antibiotic cocktails at sub-MIC levels for the treatment of multidrug-resistant *P. aeruginosa* infections.

## 1. Introduction

Antibiotic resistance is currently considered one of the biggest threats to global health [1]. Without immediate action, we are fast approaching a post-antibiotic era in which infections caused by multiple-drug-resistant (MDR) bacteria could kill 10 million people per year by 2050 [2]. Recently, the WHO released a list of 12 antibiotic-resistant bacteria that are the greatest threat to human health. Among them, three are listed as a critical priority for research and the development of new antibiotics, including *Pseudomonas aeruginosa* [3]. This bacterium is a leading cause of hospital-acquired infections (HAI), specifically in severe burn injuries, which are among the most destructive forms of trauma [4]. Infections of burn wounds represent a major concern, as infections by MDR bacteria such as *P. aeruginosa* are often life-threatening and hard to cure, due to antibiotic resistance [5]. Standard care for wound infections includes surgical debridement in combination with a single antibiotic therapy [6,7]. Dual therapy with a beta-lactam and an aminoglycoside or a fluoroquinolone is also suggested for high-risk patients, to increase the likelihood of treatment success and decrease the risk of selection of resistance [6,7].

Bacterial viruses, or phages, are considered the most abundant life form on Earth, with estimates ranging up to 10^31^ particles [8,9]. Unlike antibiotics, phages have the advantage of being highly specific, meaning beneficial bacteria will stay unharmed during treatment [10,11]. Phages are ubiquitous, and it is relatively cheap to isolate a phage that is active against an antibiotic-resistant bacterial pathogen compared to developing new antibiotics [12,13,14]. Phages have been in clinical use in Eastern Europe since the 1930s, and there has recently been increasing global acceptance of their potential as antibacterial agents [15]. As a result, we are now witnessing a rise in the number of phage-based antibacterial products that are either approved for use or undergoing evaluation [16]. However, the few clinical trials that have been conducted to date have shown variable results [12,13]. This is largely due to the many unknowns of phage pharmacology [12,13]. Unlike antibiotics, the administration of high doses of phages is not feasible due to their immunogenicity [12,17]. Thus, treatment success relies on phage replication in the presence of the target host [8,13], and phage replication should exceed bacterial replication to achieve a viable antibacterial effect in a clinical setting [12,18]. Phages vary highly in their infection properties and are not equally effective against a target bacterium; thus requiring extensive experimental studies investigating the efficacy of phages and their pharmacodynamics before reaching a clinical trial [13,19].

One limitation of using phages as antibacterial agents is that the isolation of phages that efficiently and rapidly kill their bacterial host has sometimes proven difficult [12,20]. Temperate phages, which integrate into their bacterial hosts without causing immediate lysis, and phages with low adsorption rates or an extended latent period are suggested to be less effective and not suitable for phage therapy [13,20]. However, improving the efficacy of phages with lower killing efficiency against multi-resistant bacteria is possible. For example, combinations of temperate phages have therapeutic potential [21]; and forcing phages into the lytic replication cycle by deleting genes responsible for lysogeny and integration has also been used [22]. Alternatively, combining individual lytic phages or using a phage–antibiotic combination can also improve the efficiency of the phage therapy [23,24].

Combinations of phages and antibiotics can enhance bacterial pathogen suppression, improve phage and/or antibiotic diffusion through biofilms, and reduce risks of development of resistance [25]. Phages can induce an evolutionary trade-off in multi-drug-resistant bacteria, leading to increased antibiotic sensitivity in phage-resistant mutants [26,27]. By selecting phages that bind to multidrug efflux pumps, it is possible to restore sensitivity to several antibiotic classes [26,27]. Phages can also be paired with antibiotics that inhibit bacterial processes not needed for phage production. For example, combining ϕKZ phages with antibiotics that inhibit bacterial RNA polymerases (RNAPs), such as rifampin, should be synergistic, as ϕKZ phages encode their own RNAP and are thus not dependent on the bacterial host transcription for replication [28]. Synergistic interactions between *Caudovirales* phages and several antibiotics at different concentrations, including ceftazidime, ciprofloxacin, piperacillin, and meropenem, have been reported in multiple studies [24,25,29]. Yet, phage–antibiotic interactions are not always synergistic, and antagonistic and naturalistic interactions have also been described [24,25,30]. For example, bacteriostatic antibiotics such as tetracycline, which inhibit protein synthesis in bacteria, can interfere with phage production [25]. Thus, phages and antibiotics must be carefully selected when considering combination therapies.

Most studies have examined the interactions between one phage and antibiotics at concentrations above the minimum inhibitory concentration (MIC) level. However, high doses of antibiotics can be toxic to humans [31]; antibiotics can be antagonistic to the activity of phages at concentrations equal to, or over, the MIC [23]; and recent studies have shown that low doses of antibiotics can be effective in clearing bacteria [32]. Furthermore, these studies have mostly used partially-characterized phages, thus missing critical information about the infection properties of these phages, their host range, and the stability of these phages in care products used to treat wound infections [33,34,35].

Here, we isolated 16 *Pseudomonas* phages from different environmental sources and selected three based on their performance in the efficiency of plating (EOP) test for further analysis. We characterized the genomic, phylogenetic, and morphological features of the selected three phages, as well as their infection properties and their stability under different temperatures, pH values, and in two standard burn ointments. We finally tested the efficacy of these phages against a *P. aeruginosa* strain isolated from a burn patient under multiple treatment scenarios—single phages, a mixture of two phages, and a combination of phages and antibiotics at sub-MIC and MIC levels—and determined if phage resistance had developed in treatments. We demonstrated that these phages enhanced the efficacy of the antibiotics used, even at subinhibitory concentrations, against MDR *P. aeruginosa*. Our work provides support for the potential of using combinations of two phages and one antibiotic at sub-MIC and MIC levels.

## 2. Materials and Methods

### 2.1. P. aeruginosa Strains

Twelve different MDR *P. aeruginosa* strains that were previously isolated from burn wound patients were selected for phage isolation [36]. We chose *P. aeruginosa* isolate #14 for our further phage–bacterium interactions study as it was showing a high-level of antibiotic resistance and susceptibility to all three selected phages. This study was conducted following the Institutional Review Board approved studies: IR.UMSHA.REC.1396.923, Hamadan University of Medical Sciences.

### 2.2. Minimum Inhibitory Concentration

We selected four antibiotics of different classes—ciprofloxacin, gentamicin, colistin, and imipenem—commonly used to treat *P. aeruginosa* for further analysis [6]. The MICs of *P*. *aeruginosa* isolate #14 to these antibiotics were determined using the microdilution method [37]. The estimated MICs of the antibiotics for *P. aeruginosa* isolate #14 were 128 μg·mL^−1^ for ciprofloxacin, 256 μg·mL^−1^ for gentamicin, 0.75 μg·mL^−1^ for colistin, and 2 μg·mL^−1^ for imipenem. From these four, we chose two antibiotics—ciprofloxacin and gentamicin—that are more available, affordable, and less toxic, especially when compared to colistin [38,39,40].

### 2.3. Isolation of Phages

To increase the chance of isolating phages with high genetic diversity, we sampled three different environmental sources—hospital sewage, a local river, and soil (Rasht: 37° 16′ 51″ N, 49° 34′ 59″ E and Hamadan: 34° 48′ 0″ N, 48° 31′ 0″ E ). The phages were isolated by mixing 50 mL of the collected water (soils were mixed with phosphate-buffered saline (PBS) at a ratio of 1 to 10 weight to volume (*w*/*v*)) with 50 mL of double strength LB and 10 mL of an overnight culture of each target bacterial strain. After 18 h of incubation at 37 °C, 10 mL of the mixture was centrifuged at 4000× *g* for 15 min, and sterile-filtered through a 0.45-μm membrane filter [41,42]. The filtered phage lysates were checked for phages using a standard plaque assay [43]. The filtrates were diluted in LB at five different dilutions, from 10^5^ to 10^9^. Then, 100 μL of diluted phages and 200 μL of target bacteria were mixed with 2.5 mL soft agar, spread on LB agar plates, and incubated overnight at 37 °C [41]. The harvested phages were selected according to their plaque morphology. Phages that showed clear plaques (non-turbid), as a sign to differentiate lytic phages from temperate, were selected and re-isolated by plaque purification from the LB agar plates when multiple phages on the same plate were suspected. Isolated phages were kept in LB at +4 °C for further characterization [41,44].

### 2.4. Storage and Stability of Phages

#### 2.4.1. Temperature and pH

The stability of phages at different temperatures was determined by incubating phages at the same concentration at different temperatures (30 °C, 37 °C, 40 °C, 50 °C, 60 °C, and 70 °C) for 1 h. The effect of different pH values (3, 4, 5, 6, 7, 8, 9, 10, and 11) on the activity of phages was studied by mixing 1 mL of phage with 9 mL of PBS, adjusted at different pH values. The viability of phages was tested after a 24-h incubation at 37 °C, using plaque assays according to Grygorcewicz et al. [45].

#### 2.4.2. Burn Ointments

Phages at a concentration of 10^10^ pfu/mL were mixed at a ratio of 1:1 *w*/*v* with two standard topical antimicrobial ointments for burn wounds: (1) nitrofurazone that contains nitrofurazone 2%, polyethylene glycol 300, polyethylene glycol 1000, and polyethylene glycol 3000; and (2) silver sulphadiazine that consists of silver sulfadiazine 1%, polysorbate 60, polysorbate 80, glyceryl monostearate, cetyl alcohol, liquid paraffin, propylene glycol, and water [35]. The pH values of these two products are around 6. Nitrofurazone and silver sulphadiazine are the two active compounds of these ointments, with antibacterial and antifungal activity [35]. The phage-containing ointment mixtures were incubated at 37 °C for 2, 4, and 24 h, and the stability of phages was quantified by plaque assay after each time interval. The effective therapeutic titer (ETT) was set at 10^7^ pfu/mL, according to Merabishvilli et al. [35].

### 2.5. Single-Step Growth Curve

A culture of *P*. *aeruginosa* isolate #14 in mid-exponential phase was infected by phages 6, 32, and 45 at an MOI of 0.1 and incubated with shaking at 37 °C. Then, 500 μL of sample were collected every 10 min, centrifuged, filter-sterilized through a 0.45-μm membrane filter, and kept on ice until titration. For each time point, free phages were counted using a plaque assay. Burst sizes were calculated by dividing the average phage titers of the time points after the burst from the initial average of infecting phage titers [46].

### 2.6. Adsorption Degree

The host bacterium *P*. *aeruginosa* isolate #14 was grown to mid-exponential phase, and phages added at a concentration approximately equivalent to a multiplicity of infection (MOI) of five. After five minutes of incubation at 37 °C, the culture was centrifuged, filter-sterilized, and kept on ice until titration. The adsorption degree was calculated by subtracting the number of adsorbed phages after five minutes relative to the phage concentration initially added to the culture.

### 2.7. Phage–Antibiotic Combination

Three phages at a different initial multiplicity of infection (MOI) (0.01, 0.1, 1, and 10) were mixed with *P. aeruginosa* isolate #14 (~10^8^ cfu/mL) and incubated at 37 °C. The enumeration of phages and bacterial cells at time points 60, 120, 240, and 300 min was performed using plate counts (CFU and PFU). For the phage–antibiotic combination conditions, phages and bacterial hosts were mixed at an MOI of 1 with subinhibitory concentrations (1/4 MIC) of gentamicin and ciprofloxacin, and phage and bacterial counts were determined every 60 min for 420 min. We then determined the efficacy of the selected two-phage cocktail at a combined MOI of 1 in combination with the antibiotics at sub-inhibitory concentrations (1/4 MIC) against *P. aeruginosa* for an extended period of time, at 4, 8, 12, 24, and 48 h.

### 2.8. DNA Extraction and Sequencing

Phages were enriched using the bacterial culture in the exponential growth phase, and were grown at 37 °C on a shaker incubator for 4–6 h. Phages and bacteria were separated by 30 min of 6000× *g* centrifugation. The phage supernatants were filtered through a 0.45-μm syringe filter to remove the remaining bacteria [11]. The filtrates were centrifuged at 35,000× *g* for 2 h to collect the phages in the pellets. The phage pellets were resuspended in 1 mL of SM buffer (NaCl/MgSO_4_•7H_2_O/Tris-Cl/H_2_O). The resuspended pellets were treated with DNase I for 2 h at 37 °C to remove non-phage-derived DNA. DNase I was inactivated by heating the suspension at 65 °C for 30 min. Ten μL of 20% SDS and 40 μL of protease K (20 mg/mL) were added to each 500 μL of sample, and incubated for 1 h at 37 °C. Subsequently, 35 μL of 5M NaCl and 28 μL of 10% cetrimonium bromide /0.7 M NaCl were added to the mixture and incubated at 65 °C for 30 min [11]. The aqueous phase was transferred to a new tube, 95% ethanol was added up to two times of the total volume, and samples were incubated overnight at −80 °C. Tubes were then centrifuged at 16,000× *g* at 4 °C for 1 h [11]. Supernatants were removed, and DNA pellets were resuspended in 100 μL of Tris-EDTA buffer. The extracted DNA was purified using the Genomic DNA Clean & Concentrator, ZYMO Research (Cat No./ID: D4064), according to the manufacturer’s protocol. The concentration of the purified phage DNA was measured via Qubit 3 [11,44]. Isolated phage DNA was sequenced using an Illumina MiSeq PE250 at the McGill University and Genome Quebec Innovation Center (http://gqinnovationcenter.com/index.aspx accessed on 1 January 2021).

Bacterial DNA was extracted using DNeasy Powersoil Kit, Qiagen (Cat No./ID: 12888-100) according to the manufacturer’s protocol. Isolated bacterial DNA was sequenced using an Illumina MiSeq PE250 at the McGill University and Genome Quebec Innovation Center (http://gqinnovationcenter.com/index.aspx accessed on 1 January 2021).

### 2.9. Genome Assembly and Annotation

Raw reads produced by sequencing were trimmed for the adapters and quality checked using BBDuk (http://jgi.doe.gov/data-and-tools/bb-tools/ accessed on 1 January 2021); default options were used. Trimmed reads were subsequently merged by FLASH [47]. Dedupe (https://docs.dedupe.io/en/latest/ accessed on 1 January 2021) from BBTools was used to remove duplicate reads. SPAdes (https://github.com/ablab/spades accessed on 1 January 2021) was used for the de novo assembly. ORFs were identified for the complete genome using Glimmer (https://ccb.jhu.edu/software/glimmer/ accessed on 1 January 2021) [48]. Identified ORFs were annotated using BlastX (E-value < 1 × 10^−5^). The analysis was performed within Geneious Prime. The phage genomes have been submitted to NCBI and can be found under the GenBank accession numbers MN563783, MN563784 and MN563785; the bacterium sequence data can be found under the accession number PRJNA595666.

### 2.10. Phylogenetic Analyses

The phylogenetic analyses of amino acid sequences from the major capsid proteins were selected from three isolated phages to construct individual trees. Similar proteins were identified using a blast search with a minimum of 40% sequence identity, over at least 90% of the sequence length. Three conserved phage genes (two structural proteins, the head protein and tail protein, and the phage DNA-packaging protein, the terminase large subunit) were combined to compare similarities among the three isolated phages. Phylogenetic analyses were performed by using the maximum likelihood (ML) method based on the Whelan and Goldman model [49], with 500 bootstrap replicates. The analysis was performed within MEGA. We used progressiveMauve [50] for the DNA level homology comparison between phages.

### 2.11. Transmission Electron Microscopy (TEM)

Filtered phage lysates were centrifuged for 2 h at 25,000× *g*, and phage pellets were resuspended in PBS. Then, 10 μL of resuspended phage pellets were deposited on carbon-coated grids for 10 min and stained with 2% uranyl acetate. The negatively-stained grids were then observed on a Leo 906 E microscope (Carl Zeiss, Germany) at 80 kV.

## 3. Results

### 3.1. Phage Isolation and Selection

A total of 16 lytic phages were isolated from different sources, including soil (three phages), hospital sewage (four phages), and a local river (nine phages). We then selected four phages based on their performance in an initial EOP test (Appendix A). The selected phages were further genetically characterized. We eventually selected three phages with distinct genetic features and infection properties out of the four initially sequenced phages (Appendix A). All of the selected phages were able to form plaques on *P. aeruginosa* isolate #14, although none of these phages were originally isolated against this isolate, as per the ready-to-use approach recommended for phage therapy. Phages used in this study were isolated using *P. aeruginosa* isolates 6, 32, and 45. They were named according to the bacterial viruses guidelines [51]: vB_PaeM_GUMS6, vB_PaeM_GUMS32, and vB_PaeM_GUMS45, but only the last part of these names is used in this paper for simplicity (i.e., 6, 32, and 45).

### 3.2. Phage Stability under Different Temperatures, pH Conditions, and in Burn Ointments

We then assessed the stability of all three phages in two commonly-used burn ointments, nitrofurazone and silver sulphadiazine, for 24 h. Isolated phages were quantified using plaque assays after 2-, 4-, and 24-h incubations with the burn ointments at 37 °C. We observed a significant loss of all phages during the incubation with the ointments (Figure 1A). We further tested the survival of the selected phages under different temperatures and pH values. The three selected phages were most unstable at 60 and 70 °C, and acidic pH values of 3 and 4 (Figure 1B).

### 3.3. Genome Characterization, Annotation, DNA Homology, Phylogeny, and Morphology

De novo assembly of the resulting 339,932 reads of phage 6 resulted in a 66,806 bp genome (G + C content: 55.1%) with a high similarity at the nucleotide level (about 93% pairwise identity over 79% coverage) to the genome of vB_PaeM_C1-14_Ab28, a related phage (Figure 2A and Appendix A). A total of 400,544 reads were produced for phage 32, which resulted in a 280,024 bp genome (G + C content: 36.9%) after assembly using SPAdes (Figure 2A). The genome showed resemblance to *Pseudomonas* phage phiKZ—a jumbo phage—[28] with 94.4% pairwise identity over 54% coverage (Appendix A). For phage 45, a genome of 92,218 bp (G + C content: 52.8%) was constructed via the assembly of 346,714 reads; this genome is most similar to vB_PaeM_C2-10_Ab02 (97% pairwise identity over 58% coverage) (Figure 2A and Appendix A). In addition, we screened the genome of *P. aeruginosa* isolate #14 for prophages. Contigs longer than 10 kb, produced by MIRA, were submitted to the PHAST web server (http://phast.wishartlab.com accessed on 1 January 2021) to predict potential prophages. Four prophages were identified, three intact and one questionable.

The three phages were separated into different phylogenetic clades constructed using the major capsid protein. Phage 6 was the most similar to Pseudomonas phages R26 and PB1 for the major capsid protein. Phage 32 was related most to phages phiKZ, and fnug. Phage 45 was most related to phages PA10 and Delftia (Figure 2B–D). A single phylogenetic tree was also constructed using two structural proteins (major capsid and tail fiber) and one functional protein (terminase large subunit) to compare the similarity between phages 6, 32, and 45. Phages 32 and 45 were more similar compared to phage 6 (Figure 2E). The isolated phages can be divided into the following phage genus, according to ICTV classification: Pbunavirus, phage 6; Phikzvirus, phage 32; and Pakpunavirus, phage 45.

The transmission electron microscopy analysis revealed a *Myoviridae* morphology for all three phages. Phage 6 was the smallest phage, with an icosahedral head of approximately 76 × 76 nm connected to a 170 nm tail; and phage 32 was the biggest phage, with an icosahedral head of approximately 166 × 150 nm and a 250 nm long tail. Phage 45 possessed a head with dimensions of 156 × 125 nm attached to a 234 nm long tail (Figure 3). Phage 6 was shown to have a latency period of 60 min and the smallest burst size of 6.9 ± 0.5. Phage 32 had the longest latency period (100 min) and a burst size of 14 ± 2.9. Phage 45 possessed the shortest latency period (50 min), and the biggest burst size of 250 ± 107.

### 3.4. In Vitro Phage Efficacy

We selected three phages with the highest genetic differences and distinct infection properties (Figure 2, Figure 4A and Appendix A) out of the four sequenced ones for the in vitro efficacy test. We first tested the activity of each individual phage against *P. aeruginosa* isolate #14 using four different multiplicities of infection (MOIs) ranging from 0.01 to 10, and including the initial MOI, as phages showed an adsorption degree higher than 95%). Phage 6 showed relatively higher efficiency against *P. aeruginosa* isolate #14 and was successful in lowering the concentration of the target bacteria more efficiently after 4 h of infection (Figure 4B–D). The reduction level was significant only when phage 6 was applied at an MOI of 10 after 4 h of infection (Figure 4B–D).

To test whether the application of a phage cocktail could improve the inhibitory effect of phages within 5 h of infection and avoid bacterial regrowth, we added the second phage (32 or 45) 4 h after the first phage. We calculated an MOI of 1 for the second phage, based on the bacterial concentration at the time of administration (Figure 4E), to minimize possible side effects. We also examined whether a combination of phage 6 and two conventional antibiotics (gentamicin and ciprofloxacin, at a 1/4 of MIC) could promote the efficiency of the antibiotic treatment (Figure 4F). We selected these antibiotics because of their lower toxicity and higher availability. Our results show that treatment outcomes differ based on the type of phage or antibiotic used in combination with Phage 6. We further tested potential synergistic effects between two phages and one antibiotic against *P. aeruginosa* isolate #14 over an extended 48-hour period. All tested treatment scenarios significantly decreased the concentration of the targeted bacterium by more than one log (Figure 5A,B). We observed the highest efficacy when phages 6 and 45 were used in combination with gentamicin, lowering *P. aeruginosa* isolate #14 concentration up to 3 logs after 12 h (see Figure 5A,B). However, this was followed by a rise of phage-resistant phenotypes. Phage concentrations significantly decreased over time in all treatment scenarios (Figure 5C). We also tested the effect of using lower initial concentrations of *P. aeruginosa* isolate #14 on the efficacy of the treatment under the same conditions. We treated *P. aeruginosa* isolate #14 with phage–antibiotic combinations at two different initial concentrations—10^6^ and 10^7^ CFU·mL^−1^. We observed a similar bacterial killing effect with a 2 to 3 log decrease after 12 h when compared to the higher initial concentration (Figure 5D). In contrast, we observed a higher efficacy when phages and gentamicin were co-incubated with *P. aeruginosa* isolate #14 at the early- and mid-exponential phase compared to the late-exponential phase (Figure 5E).

To see whether resistance developed before 24 h, we sampled two additional time points, at 8 and 12 h. We observed that resistance developed after ~12 h of co-incubation of bacteria and phages. We randomly selected a total of 120 bacterial colonies, from time points 12 and 24 h, to test their susceptibility to the combination of phages and gentamicin. For the phage 6 + phage 32 + gentamicin combination, 50% of the selected colonies were still susceptible to one of the two phages after 12 h, in contrast to only 30% of colonies for the phage 6 + phage 45 + gentamicin combination. All the selected colonies for the 24-h time point were resistant to all three phages (Figure 5F). We did not detect any changes in the MIC for the selected colonies for either time point. Similar results were observed when we used the MIC concentration. To test whether antibiotic treatment could induce infectious prophages that were predicted in silico, we exposed *P. aeruginosa* isolate #14 to antibiotics (gentamicin and ciprofloxacin at 1/4 and the MIC). After 1, 2, 4, 24, and 48 h of induction with the different antibiotics at said concentrations, we collected phage supernatants and proceeded with plaque assays on 10 other *P. aeruginosa* strains (Appendix A). We detected more infectious prophages after 2 and 4 h of exposure to antibiotics, respectively. The final MIC did not seem to impact prophage induction, as no clear differences were observed between 1/4 and the MIC (Appendix A).

## 4. Discussion

The increasing incidence of infections caused by MDR bacteria is a global health concern. *P. aeruginosa*, a leading cause of nosocomial infections, is one of the three bacterial species in the WHO’s list of critical pathogens [2,52]. Phages might provide a better alternative to antibiotics due to their specificity and relative ease of isolation [53]. In phage therapy, multiple phages are usually applied at once as a cocktail to broaden the spectrum of bacterial targets, prevent resistance, and increase efficacy under a collective killing effect [12]. Importantly, phage therapy is not limited to using only phage cocktails, as recent studies have reported that combining phages with antibiotics can also be synergistic [24]. In addition to increasing efficiency, the phage–antibiotic combinations may prevent the development of bacterial resistance and enhance susceptibility to antibiotics [27,54,55]. They can also lead to more acceptance of phage therapy by the medical community. However, despite the huge potential to improve phage therapy success, the synergistic interactions taking place in phage–antibiotic combinations and between phages remain relatively unexplored [23].

Here, we studied the synergetic effects of different phage–phage and phage–antibiotic combinations against a *P. aeruginosa* strain isolated from a burn wound patient. Consistent with the previous reports [24,56,57], our data indicate synergy between phages, as well as phage–antibiotic combinations against multi-resistant *P. aeruginosa,* even at sub-MIC and MIC levels. If efficient, combining phages with low doses of antibiotics for infection treatment could help reduce the risk of side effects by antibiotics and be more cost-effective for chronic infections and less antagonistic to the activity of phages [23,31,32]. Yet, caution is warranted as the usage of antibiotics at lower doses without phages has led increased bacterial resistance to antibiotics in vitro [58].

The efficacy of the treatments varied according to the phage-to-bacteria ratio, but generally phages showed higher efficiency at higher MOIs. This is consistent with previous studies showing a positive correlation between high doses of phages (10^7^–10^9^ pfu) [13,59], and the efficacy of the treatment [60]. However, the administration of phages at high doses can be a double-edged sword as a high concentration of phages is known to activate the host’s immune response, which can lead to the neutralization of the administered phages [61]. We therefore sought to determine whether the bacterial killing effect of phages could be enhanced at lower doses by combining them with other phages and/or antibiotics. We evaluated the killing efficiency of two-phage cocktails, one-phage–antibiotic, and two-phage–antibiotic combinations against one strain of MDR *P. aeruginosa*. Our data show the highest efficacy for simultaneous application of two-phages–one-antibiotic combinations against the *P. aeruginosa* isolate #14. This is likely due to a high fitness cost encumbered by the bacterial pathogen associated with fighting at multiple fronts, as it has been suggested that coping with multiple stressors can negatively affect bacterial growth [62,63].

As with antibiotics, bacteria can become resistant to phages [64]. One way to limit the development of resistance is to use phage cocktails that include genetically-distinct phages [33,65]. We therefore aimed to determine if a cocktail of genetically-distinct phages, with or without antibiotics, could overcome the development of resistance in *P. aeruginosa* isolate #14. We observed the survival of a resistant subpopulation after 12 h, which expanded after 24 h. Our findings are in agreement with previous studies that reported a time window of between 6 and 48 h after phage infection for the rise of phage-resistant phenotypes [55,64,66]. Resistance to multiple phages may need several resistance mutations to occur, which increases the associated fitness costs [63]. We tested if resistance to one phage could protect the bacterium against infection by other phages. We found that the resistant colonies isolated from the co-incubation of *P. aeruginosa* isolate #14 with one of the phages were also resistant to the other phages in the study. This suggests all three phages use a common receptor to infect the bacterial host, and that resistance could have been caused by mutations in the phage receptor on the bacterial surface [67]. Further molecular analyses of resistance phenotypes are warranted to draw a clear picture of the resistant mechanisms used by *P. aeruginosa* isolate #14 against phages or antibiotics.

The reestablishment of antibiotic sensitivity through phage infection has been previously reported [27]. It has been suggested that phage selection can produce an evolutionary trade-off in multi-resistant bacteria, resulting in increased sensitivity to antibiotics [27]. Evolutionary trade-offs are often observed where organisms evolve one trait that increases fitness, while simultaneously allowing reduced performance in another trait [68]. In our study, we did not observe any changes in the susceptibility of *P. aeruginosa* isolate #14 to gentamicin after its co-incubation with phages and the antibiotic. Thus, the phages we selected failed to increase the sensitivity to the antibiotics likely due to their inability to impose an evolutionary trade-off in this short amount of time.

We further aimed to determine if the selected phages could be used along with burn-wound-care products to enhance antimicrobial pressure during treatment. Many care products are highly acidic antiseptics, which can negatively-impact phage activity [35]. We tested the stability of the phages in two common ointments (nitrofurazone and silver sulphadiazine) over 24 h. Our data show that the incubation of phages with different ointments can result in a significant phage concentration loss after only 2 h. This suggests that phage stability in standard wound care products should be carefully assessed before applying them together. Yet, the application of alginate templates [69] or newly-developed phage hydrogels [70] might provide a solution. One must also consider that most phages can be adsorbed within the first ten minutes of exposure to their host, which might be enough for phages to infect the target bacteria before being inactivated by the acidity of the care products. It should thus be possible to administer phages separately and before applying ointments to give them enough time to infect the target bacteria.

Our results still need an in vivo validation as in vitro resistance might not be relevant in vivo [71]. In a more-complex environment like the human body, bacteria must respond to different environmental stressors, which increases the fitness costs of genetic loss [45,72,73]. Therefore, the evolution of resistance to phages can be more challenging in vivo. In addition, host-associated factors such as the immune response to phages and the pathogenic bacteria targeted can strongly impact phage–bacteria interactions. However, our results highlight that the rational design of phage–antibiotic cocktails to target MDR bacteria can enhance the efficacy of treatment even if low doses of phages or antibiotics are applied. Finding ways to improve the antibacterial activity of phages with lower efficacy can expand our arsenal against multi-resistant bacteria. In addition, as antibiotics are becoming less effective against several bacterial pathogens, combining them with phages is a possible strategy to maintain their clinical effectiveness. To limit the evolution of resistance in bacteria or restore sensitivity to current antibiotics, phage cocktails should carefully be designed to impose evolutionary trade-offs in the targeted pathogen.

Going forward, we are planning to validate our results in vivo as it is highly likely that other aspects of the host, such as the immune response [74] and its structural features [75], will also affect phage–bacteria interactions. Using mouse models of wound burn [76], we will study the efficacy of our phage–antibiotic combinations as well as the underlying development of phage-resistant bacteria.

## Figures and Tables

**Figure 1 viruses-13-00334-f001:**
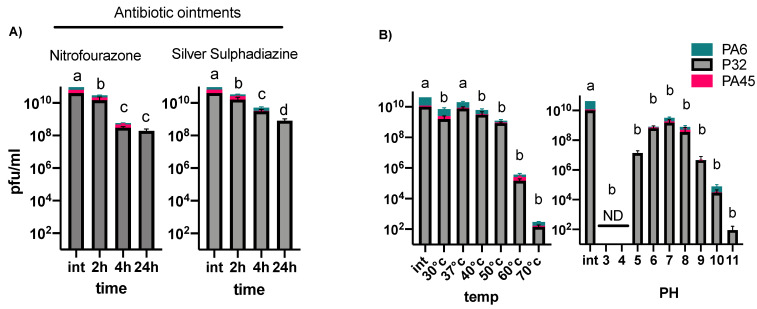
Stability of phages in two burn-wound-care products at different temperatures and pHs. (**A**) Stability of three phages, 6, 32, and 45 in nitrofurazone and silver sulphadiazine over 24 h and (**B**) stability of phages at different temperatures and pHs. Bars connected by the same letter (a, b, c and d) are not significantly different (*p* < 0.05, two-way ANOVA, Tukey’s multiple comparisons test); ND, not detected; Int, initial concentration. Error bars represents three biological replicates.

**Figure 2 viruses-13-00334-f002:**
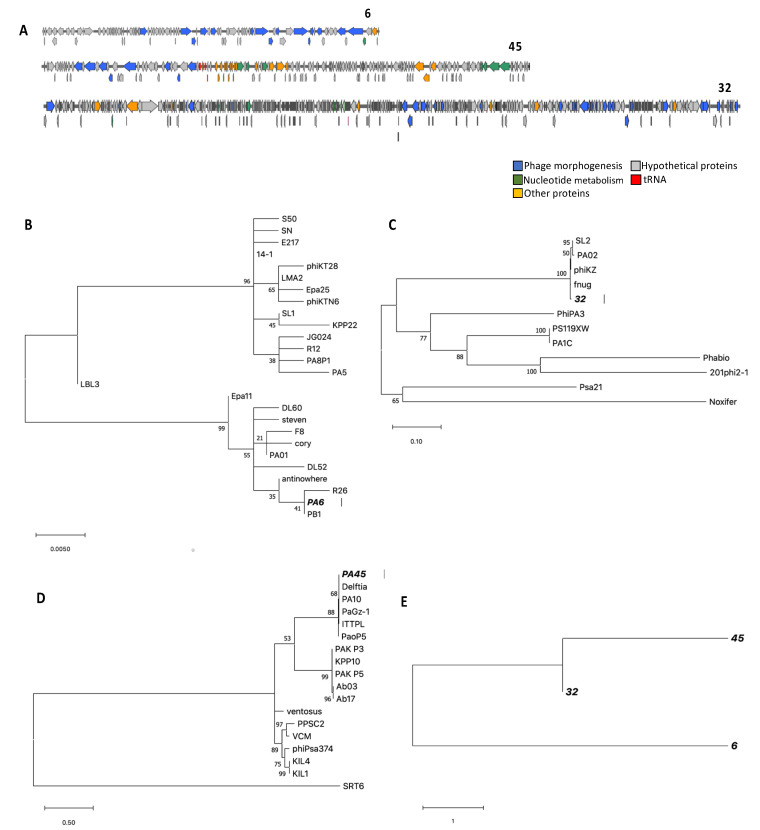
Genomes and phylogenetic trees of phages 6, 45, and 32. (**A**) The predicted coding sequences (CDSs) are indicated by arrows. CDSs predicted to encode structural proteins are indicated in blue, hypothetical proteins in grey, and nucleotide metabolism in green. Genes predicted to encode transfer RNA are indicated in red and yellow shows other proteins. (**B**–**D**) Phylogenetic tree showing the relationships of the major capsid, of three phages. (**E**) A single phylogenetic tree of head protein, tail protein, and terminase large subunit combined, of phages 6, 32, and 45. The tree was inferred by using the maximum likelihood (ML) method based on the Whelan and Goldman model [23]. The analyses were conducted in MEGA.

**Figure 3 viruses-13-00334-f003:**
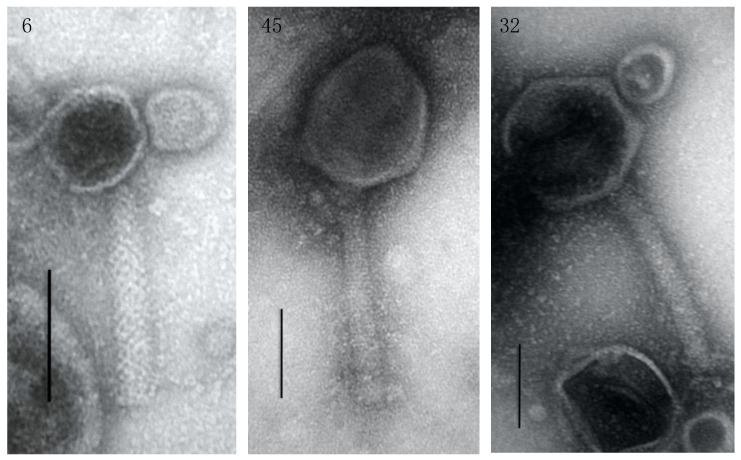
Transmission electron microscopy (TEM) of phages 6, 32, and 45. Phages were negatively stained with 2% uranyl acetate. Scale bars represent 100 nm.

**Figure 4 viruses-13-00334-f004:**
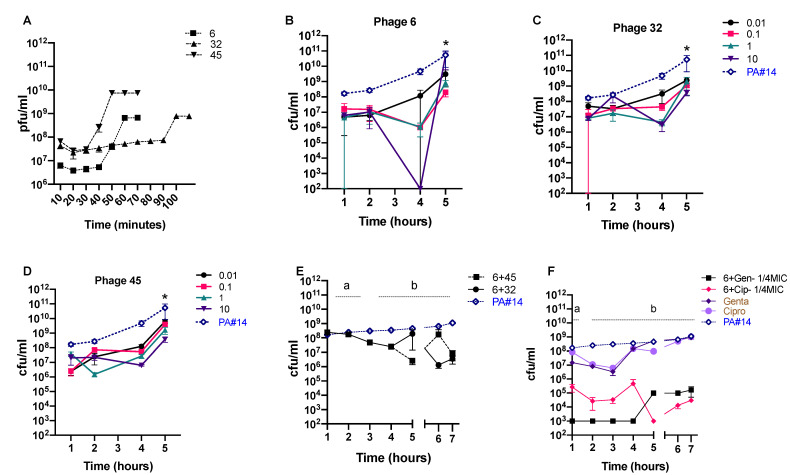
Effect of phage and antibiotic treatment in killing a *P*. *aeruginosa* strain from a wound patient. (**A**) A single-step growth curve of three phages; (**B**–**D**) efficacy of phages 6, 32, and 45 against *P. aeruginosa* isolate #14 at different multiplicities of infection (MOIs); (**E**) effect of a two-phage cocktail (6 + 32 and 6 + 45) against *P. aeruginosa* isolate #14; (**F**) effect of phage 6 in combination with two different antibiotics (gentamicin and ciprofloxacin) on *P. aeruginosa* isolate #14. PA#14, *P. aeruginosa* isolate #14. Points connected by the same letter (a or b) are not significantly different; asterisks show significant differences (*p* < 0.05, two-way ANOVA, Tukey’s multiple comparisons test). Error bars represent three biological replicates. Four different MOIs: 0.01, 0.1, 1, and 10 were used for panels B, C, and D, while a MOI of 1 was used for panels E and F. Bacteria-only controls are in blue; antibiotic + bacteria controls are in brown. Dashed lines in panels B, C, and D represent one common control.

**Figure 5 viruses-13-00334-f005:**
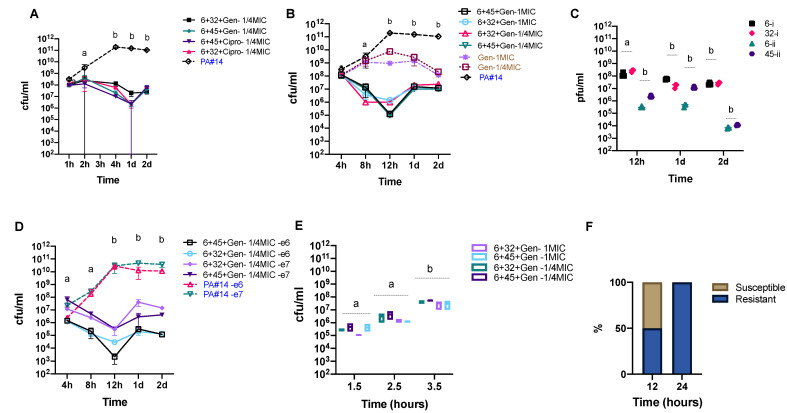
Combinations of phage and antibiotic treatments against a multi-resistant *P*. *aeruginosa* strain. (**A**,**B**) Effect of phage + antibiotic combinations on *P*. *aeruginosa* isolate #14. (**C**) Densities of phages at three different time points of the treatment. (**D**) Bacterial killing effect of phages and gentamicin 1/4 minimum inhibitory concentration (MIC) against *P. aeruginosa* isolate #14 at lower initial concentrations (10^6^ and 10^7^ cfu/mL). (**E**) Efficacy of phages against *P. aeruginosa* isolate #14 after 12 h of co-incubation and at different stages of the exponential phase (early, middle, and late). (**F**) Proportion of colonies resistant or susceptible to one or all phages, after 12 and 24 h of co-incubation with phages and antibiotics. PA #14, *P. aeruginosa* isolate #14; I, phage 6+ phage 32; and ii, phage 6+ phage 45. Points connected by the same letter are not significantly different (*p* < 0.05, two-way ANOVA, Tukey’s multiple comparisons test). Error bars represents three biological replicates. Bacteria-only controls are in blue, while the antibiotics + bacteria controls are in brown.

## Data Availability

Not applicable.

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
