# Peer review of "Improving the Inhibitory Effect of Phages against Pseudomonas aeruginosa Isolated from a Burn Patient Using a Combination of Phages and Antibiotics"

_viruses, 2021, doi:10.3390/v13020334_

Round 1
Reviewer 1 Report
Comments
- For the one step growth assay, was there a wash step to remove unadsorbed phages?
- There are no controls described in any of the experiments. This is a major oversight.
- The font used in figure 2 is too small and makes the genome and phylogenetic trees very difficult to view
- The burst size of Phage 32 should be presented as 1.4 x 1010
- It is not clear in Figure 4 what MOI was used for each graph. This needs to be made clearer
- The font in Figure S1 is again too small
- I recommend the bacterial strains be sequenced before and after phage exposure to detect what the mutations were to give rise to phage resistance
Author Response
Please See the attachment the point-by-point response to all reviewers

Reviewer 2 Report
In this research article, the authors have isolated a P. aeruginosa strain from a burn patient and they provide data upon treatment of the phage with a single phage, mixture of phages and combination of phages and antibiotics. They also provide data of the stability of the phages at different temperatures and pH and in two burn ointments. They conclude that the most efficient combination is the one of phages and antibiotics and they observe that phages are unstable at high temperatures, acidic pH and in the burn ointments. The study is well executed, well presented, is scientifically sound and has some novelty.
Specific comments for this study:
- The introduction is a bit short and mainly describes the general problem of AMR, the antibacterial properties of phages and some aspects of previous studies were phages with antibiotics are combined. The authors can expand a bit more on the specific aspects of this study and discuss about P.aeruginosa infections in burn wounds and also on the current methods of therapy of these patients as opposed to the treatment proposed here.
- The Methods section is well described and detailed. It will be beneficial if the authors provide a bit more info on the compounds of the burn ointments rather than just providing the name of those and mention some basic characteristics of these compounds, for example the pH if known or the mechanism of action.
- The Supplementary Figure S1 is not very easy to read. It will be helpful if the authors provide percentages of similarity/ homology among the phages used in the study.
- Thorough check of the minor spell mistakes is required.
Author Response
Please see the attachment for a point-by-point response to all reviewers' comments

Reviewer 3 Report
The length and organization of the ideas should be carefully considered. Below I detail some necessary but not exhaustive changes that could be done.
Abstract
The summary is poorly written. A good abstract should introduce, concisely, what is already known about the subject, what’s not known, and what is the focus of the study. Then the methods used to examine it and the main results. Finally, a conclusion with the primary message and perspectives.
Introduction
The introduction should aim at examining previous similar studies, what’s missing and what the current study adds to them. In fact, the current study dedicates a lot to the characterization of the phage, something missing in other studies, and should be highlighted and compared.
Results
Please remove some methodological details from the result section.
Discussion
the last paragraph is lacking a citation, the phage activity in the more complex environment is presented in some works: DOIs:
10.3390/ijms21124390
10.1128/AEM.02900-18
10.1089/mdr.2020.0083
These are some examples, maybe You should extend Your discussion or thing about the next study.
Methods
Too long, many methods are well known and details are not necessary (example: temperature and pH stability it should be according to ..... )
Author Response

(The authors gave the same response as above.)

Round 2
Reviewer 1 Report
The manuscript has improved significantly since reviewer's comments where addressed.
Author Response
We are happy to read that we correctly addressed the reviewer's comments, and thank the reviewer for their constructive comments.
Reviewer 3 Report
Authors correctly provide all corrections to the manuscript, at this point, manuscript will be a great piece of knowledge for further AMR combat esspecially in burn wound patients. I have one doubt to figure 4 indicated below:
Fig 4BCD - did authors prepare only one control growth curve? Control growth looks identical for all panels. maybe this is averaged from multiple experiments but should be indicated.
Author Response
Authors correctly provide all corrections to the manuscript, at this point, manuscript will be a great piece of knowledge for further AMR combat especially in burn wound patients. I have one doubt to figure 4 indicated below:
Response: We thank the reviewer for their positive feedback, and we are happy that they found our edits relevant. We also thank the reviewer for their useful comments that helped to improve this work.
Fig 4BCD - did authors prepare only one control growth curve? Control growth looks identical for all panels. maybe this is averaged from multiple experiments but should be indicated.
Response: Panels B, C, and D of figure 4 share one common control as correctly noted by the reviewer- we now indicate this clearly in the figure legend.